# An Activatable T_1_-Weighted MR Contrast Agent: A Noninvasive Tool for Tracking the Vicinal Thiol Motif of Thioredoxin in Live Cells

**DOI:** 10.3390/molecules26072018

**Published:** 2021-04-01

**Authors:** Jongeun Kang, Eunha Hwang, Hyunseung Lee, Mi Young Cho, Sanu Karan, Hak Nam Kim, Jong Seung Kim, Jonathan L. Sessler, Sankarprasad Bhuniya, Kwan Soo Hong

**Affiliations:** 1Research Center for Bioconvergence Analysis, Korea Basic Science Institute, Cheongju 28119, Korea; jekang@precision-bio.com (J.K.); hsys0307@kbsi.re.kr (H.L.); cmy78@kbsi.re.kr (M.Y.C.); sanukaran1111@kbsi.re.kr (S.K.); 2Graduate School of Analytical Science and Technology, Chungnam National University, Daejeon 34134, Korea; 3Center for Research Equipment, Korea Basic Science Institute, Cheongju 28119, Korea; hwang0131@kbsi.re.kr (E.H.); haknam81@kbsi.re.kr (H.N.K.); 4Department of Chemistry, Korea University, Seoul 02841, Korea; jongskim@korea.ac.kr; 5Department of Chemistry, The University of Texas at Austin, Austin, TX 78712-1224, USA; 6Centre for Interdisciplinary Sciences, JIS Institute of Advanced Studies and Research, JIS University, Kolkata 700091, India

**Keywords:** MR relaxivity, activatable contrast, thioredoxin, vicinal thiols

## Abstract

We have synthesized new magnetic resonance imaging (MRI) T_1_ contrast agents (CA1 and CA2) that permit the activatable recognition of the cellular vicinal thiol motifs of the protein thioredoxin. The contrast agents showed MR relaxivities typical of gadolinium complexes with a single water molecule coordinated to a Gd^3+^ center (i.e., ~4.54 mM^−1^s^−1^) for both CA1 and CA2 at 60 MHz. The contrast agent CA1 showed a ~140% relaxivity enhancement in the presence of thioredoxin, a finding attributed to a reduction in the flexibility of the molecule after binding to thioredoxin. Support for this rationale, as opposed to one based on preferential binding, came from ^1^H-^15^N-HSQC NMR spectral studies; these revealed that the binding affinities toward thioredoxin were almost the same for both CA1 and CA2. In the case of CA1, T_1_-weighted phantom images of cancer cells (MCF-7, A549) could be generated based on the expression of thioredoxin. We further confirmed thioredoxin expression-dependent changes in the T_1_-weighted contrast via knockdown of the expression of the thioredoxin using siRNA-transfected MCF-7 cells. The nontoxic nature of CA1, coupled with its relaxivity features, leads us to suggest that it constitutes a first-in-class MRI T_1_ contrast agent that allows for the facile and noninvasive monitoring of vicinal thiol protein motif expression in live cells.

## 1. Introduction

Molecular probes have a time-honored role in biology and are currently being used to understand molecular processes in complex physiological environments and to probe a range of cellular processes [1,2]. In essence, these probes act as molecular spies that allow the function of the small molecule, enzymes, coenzymes, proteins, etc., to be followed readily within a given biological locus. Over the past few decades, various fluorogenic probes have been developed to study in vitro cellular milieus [3]. However, cytotoxicity remains a major concern; in addition, many of the existing optical-based probes are unable to provide useful information in vivo. In contrast, probes that exploit magnetic resonance imaging (MRI) have a number of advantages. MRI is a noninvasive modality that can provide images with superb spatial resolution [4]. Indeed, Gd^3+^ ion-based chelating agents, a classic paramagnetic species, are used in the clinic to improve the contrast and sensitivity of MR imaging [5,6].

Over the last decade, a major research focus has centered on the development of bioresponsive contrast agents [7,8,9,10,11,12,13]. As a general rule, these probes exploit changes in the relaxation rates of the protons in water molecules coordinated to a Gd^3+^ ion as a function of a chemical reaction, a change in environment, an enzymatic process, etc. The relaxivity is mainly controlled by the number of coordinated water molecules (q), the exchange rate of the inner sphere water (k_ex_ or 1/τ_M_), and the rotational correlation time of the complex (τ_R_) [14,15]. Fast motion of the contrast agents has a negative influence on its relaxivity. Conversely, reducing its motion can lead to enhancements in relaxivity and improvements in the MRI contrast. The so-called receptor-induced magnetization enhancement (RIME) effect [16,17] is a crucial strategy that has been exploited to restrict the movement of a contrast agent and thereby increases the τ_R_ and the relaxivity. Based on this strategy, Meade et al. developed MR contrast agents that provided for a 5.8-fold increase in relaxivity upon binding to the halo tag protein [18]. Caravan et al. developed a peptide based multicenter Gd^3+^-based contrast agent for collagen that could be used to successfully image fibrosis [19]. Later, this latter team reported several elegant MR contrast agents that could be used to detect acute thrombosis [20,21]. In one case, the T_1_-relaxivity was enhanced by ~70% upon binding to the target fibrin protein [22]. Here we report two new MRI T_1_ contrast agents (CA1 and CA2) that were designed to elicit the activatable recognition of the cellular vicinal thiol motifs within thioredoxin (Trx).

Trx constitutes a class of small redox protein known to be present in all organisms. It is characterized inter alia by the presence of a vicinal thiol–motif. Trx plays a significant role in many important biological processes, including redox signaling. Thiol motifs are also intimately associated, in a redox sense, with the disulfide linkages present in a number of enzymes, such as ribonucleotide reductase [23], methionine sulfoxide reductases [24], and peroxiredoxins [25]. Dysfunctions in this class of proteins are correlated with many diseases, including cancer [26,27], diabetes [28], human immunodeficiency virus type 1 (HIV-1) [29,30,31], and neurodegenerative disease [32]. It is likely that the ability to noninvasively track the presence and fate of the vicinal thiol–motif in Trx at the cellular level will allow for an increased understanding of these related disorders while providing critical insights into Trx expression. This, in turn, might allow the early adoption of appropriate treatment protocols. Based on this appreciation, we sought to develop, and wish to report here, the T_1_-weighted (T_1_-W) MRI contrast agents CA1 and CA2, designed to validate Trx expression in live cells by providing noninvasive bright-contrast images. The synthesis, relaxivity features, and uses of these new probes for live cell MR imaging based on Trx expression levels are detailed below.

## 2. Results and Discussion

The contrast agents, CA1 and CA2, were prepared in accordance with the synthetic route depicted in Scheme 1. The vicinal dithiol binding motif *p*-aminophenyldithioarsolane [33] was incorporated into the present contrast agents to enhance their T_1_ relaxivity (r_1_) through an RIME effect. To obtain ligand 2, *p*-aminophenyl-dithioarsolane was induced to react with the diethylenetriaminepentaacetic acid (DTPA) analogue C to produce intermediate 1 via an amide type coupling reaction. Then, intermediate 1 was treated with trifluoroacetic acid (TFA) to yield the free carboxylic acid ligand 2. Similarly, ligand 4 was obtained starting from B. Details of the reaction procedures are available in the experimental section. The NMR spectra, MS, HPLC, and other characterization data proved consistent with the structures proposed for new compounds and are provided in the Appendix A.

After Gd^3+^ insertion, we measured the relaxivity of the resulting contrast agents, CA1 (possessing one linking CH_2_ chain) and CA2 (possessing five CH_2_ linkers), in a 20 mM 4-(2-hydroxyethyl)-1-piperazineethanesulfonic acid (HEPES) buffer (with a pH of 7.2) at 25 °C. As shown in Figure 1, the relaxivity (r_1_) values of CA1 and CA2 were typical of those expected for monohydrated DTPA Gd^3+^ complexes, with values of 4.54 ± 0.13 and 4.58 ± 0.13 mM^−1^s^−1^ being recorded for these two agents, respectively, leading us to the conclusion that CH_2_ chain length does not contribute to the T_1_ relaxivity in a 20 mM HEPES buffer (pH 7.2). This T_1_ relaxivity value (~4.5 mM^−1^s^−1^) of our contrast agents is slightly higher than that of one of the clinically available MRI T_1_ contrast agents, Gd-DTPA (4.1 mM^−1^s^−1^ at 63.8 MHz; Magnevist, Schering, Berlin, Germany) [34], in which the difference might mainly result from additional Trx-sensing ligands of CA1 and CA2. We then evaluated the effect of the vicinal thiols-motif protein Trx and the most abundant protein human serum albumin (HSA) on the relaxivity of CA1 and CA2. As can be seen from an inspection of Figure 2a, the relaxation rate of the water protons associated with CA1 (100 µM) at 60 MHz was found to increase gradually with an increasing Trx concentration and plateaued when the Trx concentration reached ~300 µM. Specifically, the relaxation rate (R_1_) increased by ~142% (0.77 ± 0.01 to 1.09 ± 0.03 s^−1^) in the presence of 3 equiv. (*w*/*w*) of Trx. In contrast, the relaxation rate remained unaltered in the presence of HSA. This result supports our hypothesis that the contrast agent CA1 binds chemoselectively with the vicinal dithiol motif of Trx and provides assurance that this binding can operate without any potential interference from the ubiquitous blood serum protein HSA. Surprisingly, the more flexible contrast agent CA2 (characterized by a 5-CH_2_ linking chain) showed no enhancement in R_1_ values in the presence of Trx; in fact, they were possibly diminished slightly (0.77 ± 0.02 to 0.72 ± 0.03 s^−1^ at 400 μM). As for CA1, HSA produced no significant effect on relaxivity (Figure 2b). The contrasting behavior seen for CA1 and CA2 leads us to suggest that the CH_2_ tethering linker can be used to fine-tune the responsive features of the present class of MRI contrast agents. We thus sought to understand the difference between CA1 and CA2 in greater detail.

To explore whether the differences in relativity seen for CA1 and CA2 might reflect differential binding affinities to the Trx target, 2D ^1^H-^15^N HSQC NMR experiments were performed. Herein, ^15^N-labeled Trx was studied alone and in conjunction with either CA1 or CA2. Compared with the spectra of Trx alone, there were substantial chemical shift differences (representative expanded spectra are shown in Figure 3) when the contrast agents CA1 and CA2 were added to the ^15^N-labeled Trx (Figure 3). The 2D ^1^H-^15^N HSQC NMR spectra of Trx in the presence of both contrast agents revealed no significant difference between CA1 and CA2, a result we take to indicate that both agents are bound similarly. On this basis, we conclude that the observed change in the T_1_ relaxation value can be seen when CA1 is bound to Trx and reflects a close association between the gadolinium complex and Trx, which would serve to reduce the tumbling motion of the Gd-DTPA portion of the CA1 probe. This, in turn, would result in a T_1_ relaxivity enhancement, as in fact seen in the presence of Trx.

To test the suitability of CA1 as a contrast agent for in vitro cell imaging, its cytotoxicity against MCF-7 and A549 cells was evaluated. It is a truism that a prerequisite for any contrast agent contemplated for use in vitro or in vivo imaging is that it should be nontoxic and nonharmful to living organisms. As shown in Appendix A, CA1 produces little appreciable toxicity in MCF-7 and A549 cells at concentrations up to 200 µM. We therefore concluded that CA1 might be suitable for the labeling of Trx in live cells.

With such considerations in mind, we proceeded to analyze whether the contrast agent CA1 could be used to differentiate the expression level of Trx in cells by providing a brighter T_1_ contrast image, as compared to that differentiated using the control. T_1_–W MR imaging was performed in MCF-7 cells overexpressing Trx in the presence and absence of CA1 and CA2. It was found that CA1- or CA2-treated MCF-7 cells provided brighter (31% or 19% enhancement, respectively) images than did the control cells (Appendix A). As expected based on the above predictive studies, the MR images for MCF-7 cells treated with CA1 proved to be qualitatively brighter than the corresponding cells treated with CA2. Presumably, the higher R_1_ enhancement effect seen in the case of CA1 use accounts for this difference.

Next, we investigated whether the contrast agent CA1 could be used to differentiate cells with different Trx expression levels. Toward this end, two cancer cell lines (MCF-7, A549) with different expression levels of Trx were imaged. On the basis of Western blot analyses (Figure 4a) the Trx expression level in A549 is 1.55-fold higher than that in MCF-7 cells. Both cancer cells were treated with CA1, and the T_1_-W MR phantom images were obtained. As can be seen in Figure 4b,c, CA1 provided relatively brighter T_1_-W MR phantom images in the A549 cells as compared to those provided by the MCF-cells (40% vs. 16% enhancement, respectively). These findings lead us to conclude that CA1 is a unique MR contrast agent that can be used to obtain information regarding the relative expression level of Trx in live cells.

Finally, we tested whether the T_1_-W MRI signal produced by CA1 can be correlated to the Trx expression level by knocking down the Trx expression levels in MCF-7 cells. Knockdown was achieved by means of Trx-siRNA transfection, with the expression levels confirmed using Western blot analysis (Figure 5a,b). After knockdown, T_1_-W MRI phantoms were obtained for cells that were incubated in both the presence and the absence of CA1 (Figure 5c). As shown in Figure 5a, Trx-siRNA transfected MCF-7 cells provided darker T_1_-W phantom images compared with the phantom images of MCF-7 cells incubated with CA1. This finding is consistent with our core hypothesis; namely, that CA1 gives rise to bright T_1_-W phantom images in cancer cells with normal Trx expression levels because of the relatively high concentration of vicinal thiols present in this classic redox active protein.

A quantitative comparison of the T_1_-W signal intensities (cf. Figure 5d) revealed similarly low intensities for both the control MCF-7 cells and the Trx-siRNA transfected MCF-7 cells in the absence of the MR contrast agent CA1. In contrast, in the presence of CA1 (100 µM), the MR signal intensity of the MCF-7 increased significantly compared to what was seen for the untreated MCF-7 cells. Notably, the T_1_-W MR intensity of the Trx-siRNA transfected MCF-7 cells remained relatively low under identical CA1 treatment conditions. These results are taken as further evidence that CA1 is a highly sensitive MRI probe that may be used to detect the level of vicinal thiol motifs in proteins such as Trx and can be employed without any obvious interference from other entities that might be present in live cells.

## 3. Materials and Methods

### 3.1. Materials and Instrumentation

Arsanilic (TCI, Tokyo, Japan), phenyl hydrazine (TCI), 1,2-ethanedithiol (Sigma-Aldrich, Munich, Germany) diethylenetriamine (Sigma-Aldrich), phthalic anhydride (Sigma-Aldrich), benzyl chloroformate (Sigma-Aldrich), hydrazine (TCI, Tokyo, Japan), tert-butyl bromoacetate (Sigma-Aldrich), benzyl bromoacetate (Sigma-Aldrich), N-BOC 6-amino hexanoic acid (Sigma-Aldrich), Pd/C (Sigma-Aldrich), 1-(bis(dimethylamino)methylene)-1H-1,2,3-triazolo(4,5-b)pyridinium 3-oxid hexafluorophosphate azabenzotriazole tetramethy1 uronium (HATU; Carbosynth, Compton, UK), trimethylamine (TEA; Sigma-Aldrich), N,N-diisopropylethylamine (DIPEA; Sigma-Aldrich), potassium carbonate (K_2_CO_3_) (Samchun Chem., Pyungteak, Korea), acetonitrile (Samchun Chem.) dichloromethane (Samchun Chem.), and tetrahydrofuran (THF) (Sigma-Aldrich) were purchased commercially and used without further purification. Column chromatography was performed using silica gel 60 (70–230 mesh) as the stationary phase. Analytical thin layer chromatography was performed using silica gel 60 (pre-coated sheets with 0.25 mm thickness). Mass and HPLC spectra were recorded on a mass spectrometer (Synapt G2-HDMS, Waters, Manchester, UK) and an HPLC system (YL9100, Youngin, Seoul, Korea), and NMR measurements were performed on a 400 MHz spectrometer (AvanceII, Bruker, Germany) to confirm compound structures during synthesis.

### 3.2. Synthesis of 1

Compound C was synthesized according to previously reported methods [33]. To a solution of compound A in dry THF (10 mL), compound C (620 mg, 1.0 mmol), HATU (380 mg, 1.0 mmol), and DIPEA (0.15 mL) were added [35]. The reaction was then allowed to stir for 4 h. After completion of the reaction, the mixture was diluted with water and extracted with ethyl acetate (EA). The organic layer was dried over an anhydrous sodium sulfate and concentrated under reduced pressure. The crude compound was passed through a silica gel column using EA:hexanes (Hex) = 1:1 as the eluent to produce compound 1 (600 mg, 69.80%) as a white solid. ^1^H NMR (400 MHz, CDCl_3_): δ 10.12 (s, 1H); 7.67 (d, *J* = 6.73 Hz, 2H); 7.09 (d, *J* = 6.89 Hz, 2H); 3.41 (br, 12H); 3.17 (m, 2H); 2.81 (m, 4H); 2.67 (m, 4H); 1.41 (s, 36H). ^13^C NMR (100 MHz, CDCl_3_): 170.25, 139.44, 137.42, 131.14, 119.38, 80.86, 59.01, 55.94, 53.72, 51.83, 41.57, 28.07 ppm. ESI-MS *m*/*z* [M + Na]^+^ calcd 881.325, found 881.249.

### 3.3. Synthesis of 2

To a solution of compound 1 (600 mg, 0.698 mmol) in dichloromethane (DCM) (3 mL), trifluoroacetate (TFA) (22 mL) was added and a reaction was allowed to stir for 24 h. After the reaction was deemed complete, the mixture was concentrated and triturated with ether. The solid mass was dissolved in deionized water and lyophilized to produce ligand 2 as a white solid (390 mg, 88.05%). ^1^H NMR (400 MHz, DMSO-*d*_6_): δ 7.61 (m, 4H); 3.48 (br, 10H); 3.21–2.71 (br, 12H). ^13^C NMR (100 MHz, DMSO-*d*_6_): 173.04, 167.62, 139.72, 131.29, 120.09, 116.20, 65.37, 52.69, 43.01, 40.11, 39.90, 32.37, 31.35, 31.29, 31.17, 28.22 ppm. ESI-MS *m*/*z* [M + H] calcd535.075, found 635.067.

### 3.4. Synthesis of 3

Compound 3 (yield: 71.23%) was prepared from B according to the method described for 1. ^1^H NMR (400 MHz, CDCl_3_): δ7.56 (m, 4H); 3.46 (m, 10 H); 3.23 (m, 4H); 2.81 (M, 2H); 2.65 (s, 4H0; 1.57 (m, 2H), 1.41 (br, 42H). ^13^C NMR (100 MHz, CDCl_3_): 1.72.17, 170.57, 165.75, 139.66, 137.60, 131.33, 119.54, 82.37, 56.95, 56.11, 54.68, 53.98, 51.74, 49.96, 44.96, 41.69, 38.61, 37.26, 29.69, 28.11, 27.90, 25.90 ppm. ESI-MS *m*/*z* [M + Na] calcd 994.409, found 994.387.

### 3.5. Synthesis of 4

Compound 4 (yield: 91.13%) was prepared according to the method described for 2. ^1^H NMR (400 MHz, DMSO-*d*_6_):7.59 (m, 4H); 3.47 (m, 14H); 3.2 (m, 6H), 2.98 (m, 4H); 2.32 (m, 2H); 1.58 (m, 2H); 1.51 (m, 2H); 1.37 (m, 2H). ^13^C NMR (100 MHz, DMSO-*d*_6_): 173.29, 172.90, 171.97, 140.70, 137.36, 131.74, 129.10, 119.51, 119.24, 57.19, 56.28, 54.99, 41.81, 40.50, 40.11, 39.90, 39.69, 36.70, 29.09, 28.23, 26.46, 25.14 ppm. ESI-MS *m*/*z* [M + Na] calcd 748.160, found 748.162.

### 3.6. General Procedure for the Preparation of Gadolinium Complexes

Ligands 2 and 4 (0.08 mmol) were separately placed in ultrapure water (10 mL), and the resultant solutions were adjusted to ~pH 7 using sodium bicarbonate. Gadolinium chloride hexahydrate (0.078 mmol) was dissolved in 3.0 mL of ultrapure water and added to the solution of ligand 5 in three separate aliquots. After the addition of each aliquot, the pH was adjusted back to a pH ranging from 6.5–7.0 using a 0.1 M potassium carbonate solution. The solution was allowed to stir for 30 min to allow for Gd^3+^ chelation. It was then dialyzed via dialysis membrane (Spectra/Por Biotech CE MWCO 500-1000; Carl Roth, Karlsruhe, Germany) against ultrapure water overnight and lyophilized to yield the respective complexes. ESI-MS m/z (M+1) for CA1: calcd. 789.96, found 789.98; compound purity was 98.2% (from HPLC analysis), ESI- *m*/*z* [M + K]^+^ for CA2: calcd 943.96, found 943.06.

### 3.7. T_1_ Relaxivity Measurements

The longitudinal relaxation times (T_1_) of the aqueous buffer solutions of CA1 and CA2 were measured using a standard inversion–recovery pulse sequence on a bench-top NMR system (Minispec mq60; Bruker, Germany) at 60 MHz (1.41 T). In each case, five samples, whose concentrations were 0.05, 0.1, 0.2, 0.4, and 0.8 mM in Gd^3+^ at 25 °C in 20 mM HEPES buffer (pH 7.2), respectively, were prepared separately. The amount of gadolinium in the CA1 and CA2 solution samples was determined using an inductively-coupled plasma atomic emission spectrophotometer (ICP-AES) (Optima 4300DV, Perkin Elmer, Waltham, MA, USA). The ability of the proteins (Trx and human serum albumin) to modulate the longitudinal relaxivities of CA1 and CA2 were determined using T_1_ measurements per a standard inversion–recovery pulse sequence using a bench-top NMR system at 60 MHz. The concentrations of CA1 and CA2 were fixed at 100 μM and the T_1_ relaxation time was measured by varying the concentration of Trx and HSA between 0 and 700 μM.

### 3.8. Protein Preparation

To express the Trx protein, the pET32a-Trx gene was transformed into E. coli Rosetta-gami B to generate E. coli/pET32a-Trx. The E. coli cells obtained in this way were then allowed to grow at 37 °C until the OD_600_ was approximately 0.6. After decreasing the temperature to 18 °C, the protein was induced by the addition of 1 mM isopropyl β-D-1-thiogalactopyranoside (IPTG). The cells were grown for an additional 22 h at 18 °C and purified via His-tag affinity followed by thrombin cleavage and size-exclusion chromatography. For ^15^N-labeled Trx NMR samples, the E. coli cells were grown in an M9 minimal medium supplemented with ^15^NH_4_Cl as the sole nitrogen source (99% ^15^N; Cambridge Isotope Laboratories, Inc.). Cell lysates were purified using a His-tag column followed by gel filtration (HiLoad 16/600 Superdex 75 pg, GE Healthcare, Uppsala, Sweden) equilibrated with 20 mM HEPES (pH 7.5), 150 mM NaCl and 2 mM dithiothreitol (DTT).

### 3.9. NMR Spectroscopy

NMR spectroscopic measurements were performed on 0.5 mM ^15^N-labeled Trx samples in 20 mM HEPES (pH 7.5), 150 mM NaCl, and 2 mM DTT with 10% D_2_O at 25 °C on a 900 MHz NMR spectrometer (Avance II, Bruker, Germany). The 2D ^1^H-^15^N heteronuclear single quantum coherence (HSQC) spectra of the ^15^N-labeled Trx protein were measured in the presence of either CA1 or CA2. All NMR spectra were processed using Bruker Topspin 3.0.

### 3.10. MRI Phantom Imaging

MCF-7 and A549 cells were obtained from the American Type Culture Collection (ATCC, Rockville, MD, USA). For T_1_-W MR phantom imaging, MCF-7 and A549 cells were incubated with 100 µM CA1 or CA1 for 24 h at 37 °C. The medium was then carefully aspirated and washed two times with phosphate buffered saline (PBS). The cells were pelleted in 0.2 mL tubes as MRI phantoms. MR imaging was performed on a 4.7 T MRI system (Biospec 47/40, Bruker, Germany) with a quadrature birdcage RF coil (72 mm diameter) for signal transmission and reception. T_1_-W images were acquired using a multislice multiecho (MSME) MRI pulse sequence with the following parameters: TE/TR = 8.8/400 ms, a number of average of 4, a matrix size of 192 × 192, an FOV of 6 × 3 cm^2^, and a slice thickness = of 1 mm. Signal intensities in terms of T_1_ contrast from regions of interest (ROIs) were compared between CA1 and CA2 for MCF-7 and A549 cells.

### 3.11. Cell Culture and MTT Assay

MCF-7 cells were cultured in high-glucose Dulbecco’s Modified Eagle Medium (DMEM, Gibco, Grand Island, NY, USA) and A549 cells were maintained in RPMI 1640 Medium (Gibco) supplemented with 10% fetal bovine serum (FBS, Gibco) and 1% antibiotic–antimycotic (Gibco) at 37 °C in a humidified atmosphere containing 5% CO_2_. Cell viability was measured using 3-(4,5-dimethyltiazol-2-yl)-2,5-diphenyltetrazolium bromide (MTT). To test the effects of CA1, the cells were seeded at 1 × 10^4^ cells/well on a 96-well plate. The next day, the cells were incubated with predetermined concentrations of CA1 for 24 h at 37 °C. A solution of MTT (5 mg/mL) was added to each of the wells, which were then incubated for 2 h at 37 °C to let the cells form formazan crystals. The solution was removed and 100 μL DMSO was added per well to dissolve the crystals. The intensity of the absorbance at 570 nm was then recorded using a microplate reader (XMark, Bio-Rad Lab., Berkeley, CA, USA).

### 3.12. siRNA Transfection and Western Blotting

In order to knock down the Trx gene, MCF-7 cells were transfected using an electroporation system (MaxCyte STX, Gaithersburg, MD, USA). In brief, 1 × 10^7^ cells were suspended in 100 μL electroporation buffer with Trx siRNA (Santa Cruz Biotechnology, Santa Cruz, CA, USA) and control siRNA (Santa Cruz Biotechnology). The mixture was placed in a MaxCyte processing assembly and electroporated using the OPT 3 program protocol and software provided by the supplier. After electroporation, the cells were incubated in a 5% CO2 incubator for 20 min at 37 °C and treated with 10 mL of cell culture media.

For immunoblotting, the cells were lysed with a radioimmunoprecipitation assay (RIPA) lysis buffer (iNtRON Biotechnology, Korea) containing a protease inhibitor cocktail (Sigma-Aldrich) on ice. The total protein concentration was determined using a Bradford assay (Bio-Rad Lab.). Ten μg of protein was separated by 4–20% mini-precast gels (Bio-Rad Lab.) and transferred onto a membrane (Hybond PVDF; Amersham Pharmacia Biotech, Piscataway, NJ, USA). The membranes were blocked with 5% skim milk/Tris-buffered saline–Tween (TBST) and incubated with the appropriate primary antibodies and HRP-conjugated secondary antibodies. Mouse monoclonal anti-Trx, anti-β-actin, and anti-GAPDH antibodies were purchased from Santa Cruz Biotechnology (Santa Cruz, CA, USA). Protein bands were observed using enhanced chemiluminescence (ECL) detection reagents (Bio-Rad Lab.).

## 4. Conclusions

We have presented here the synthesis of two phenyl–dithioarsolane appended DTPA ligands as the basis for a new set of T_1_-W MR contrast agents (CA1 and CA2), designed to target vicinal thiol motifs in proteins. Both contrast agents showed relaxivities typical of monohydrate Gd-DTPA-based contrast agents. No relaxivity changes were seen for either agent in the presence of the omnipresent protein HSA. In contrast, a 140% increase in relaxivity (R_1_) was seen for the contrast agent CA1, bearing the shorter linker in the presence of 3 equiv. (*w*/*w*) of Trx. ^15^N-NMR spectral data provided support for the suggestion that CA1, with one linking CH_2_ chain, is more closely associated with Trx than its more flexible congener, CA2, which contains a five-CH_2_ chain. These findings are consistent with the suggestion that the shorter chain length system (CA1) binds tightly to the vicinal thiol motif present in Trx, which leads to an enhancement in relaxivity via an RIME effect. The contrast agent CA1 provided T_1_-W MR images, the intensity of which depended on the expression of Trx in cancer cells. A control experiment in Trx siRNA-transfected MCF-7 cells revealed that CA1 provides bright images upon binding with Trx. New contrast agents, such as those described here, may emerge as useful tools that can be used to resolve specific cellular abnormalities while providing critical information relating to native redox regulation and the extent of Trx expression. This, in turn, could lead to improved or more timely treatments for a number of Trx-related disorders.

## Data Availability

The data presented in this study are available on request from the corresponding author.

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
