# Peer review of "An Activatable T1-Weighted MR Contrast Agent: A Noninvasive Tool for Tracking the Vicinal Thiol Motif of Thioredoxin in Live Cells"

_molecules, 2021, doi:10.3390/molecules26072018_

Round 1

Reviewer 1 Report

Comments

  • page 1, line 16: “protein, thioredoxin” to be substituted by “protein thioredoxin”;
  • p. 1, l. 17: “relaxivities typically of gadolinium” to be substituted by “relaxivities typical of gadolinium”;
  • p. 1, l. 33: “in a complex physiological environments” to be substituted by “in complex physiological environments”;
  • p. 1, l. 40-41: “MRI is a modality that can provide noninvasive images with superb spatial resolution” to be substituted by “MRI is a noninvasive modality that can provide images with superb spatial resolution”;
  • many other typo’s, misspells etc. can be found throughout the whole manuscript;
  • the bibliography concerning CAs that exploit the RIME effect is quite obsolete;
  • p. 2, l. 81: reference 34 is written in superscript;
  • p. 2, l. 84: where is compound C from?
  • some structures in Scheme 1 show weird angles and should be re-drawn;
  • how do the Authors exclude the possible presence of free Gd3+? Also, the MWCO of the dialysis membranes should be reported;
  • according to the NMR and MS spectra in the SI, the ligands were not pure: has the purity been assessed somehow (e. g. by HPLC)?
  • it is very important to report the temperature and pH at which the relaxivity measurements relative to Figure 1 and corresponding text were carried out;
  • how do the relaxivity values obtained for C1 and C2 compare with those reported for similar CAs, e. g. Gd-DTPA?
  • how do the Authors explain the lack of formation of adducts with HSA, that actually seems to be very likely because of a hydrophobic part in both ligands?
  • a thorough relaxometric investigation (e. g. the acquisition and fitting of NMRD profiles) is strongly recommended in order to establish important parameters (such as tauM and tauR) that would support some of the conclusions reported by the Authors.

Author Response

Reviewer 1

1. page 1, line 16: “protein, thioredoxin” to be substituted by “protein thioredoxin”; p.1, , l. 17: “relaxivities typically of gadolinium” to be substituted by “relaxivities typical of gadolinium”; p.1, l. 33: “in a complex physiological environments” to be substituted by “in complex physiological environments”; p. 1, l. 40-41: “MRI is a modality that can provide noninvasive images with superb spatial resolution” to be substituted by “MRI is a noninvasive modality that can provide images with superb spatial resolution”; many other typo’s, misspells etc. can be found throughout the whole manuscript;

Reply: Thanks for these comments. Our manuscript has been revised as per these comments.

2. the bibliography concerning CAs that exploit the RIME effect is quite obsolete;

Reply: Thanks for this comment. T1 relaxivity is increased via RIME effect, and which is the main issue of our study. We revised our manuscript only with more relevant two references (#16-17).

3. p. 2, l. 81: reference 34 is written in superscript;

Reply: We have revised our manuscript.

4. p. 2, l. 84: where is compound C from?

Reply: Compound C was synthesized with previously reported method (reference #33). We have added two sentences on this issue in the 3.2 experimental section (lines 219-221 of page 8).

5. some structures in Scheme 1 show weird angles and should be re-drawn;

Reply: Thank you for this comment. We have re-drawn all structures in Scheme 1. Now all structures are maintained their bond angles according to the hybridization. We have replaced that one in our manuscript.

6. how do the Authors exclude the possible presence of free Gd3+? Also, the MWCO of the dialysis membranes should be reported;

Reply: Thank you for this comment. We have dialyzed by dialysis membrane (Spectra/Por Biotech CE MWCO 500-1000; Carl Roth, Karlsruhe, Germany) against ultrapure water overnight and lyophilized to yield the respective complex.. We have mentioned in the experimental section of revised manuscript (lines 257-260 of page 9).

7. according to the NMR and MS spectra in the SI, the ligands were not pure: has the purity been assessed somehow (e. g. by HPLC)?

Reply: Thank you for this comment. We have performed HPLC analysis (Figure S14) to detect purity of our compound. Purity of our compound is 98.2%. We have added this information in revised manuscript including Figure S14.

8. it is very important to report the temperature and pH at which the relaxivity measurements relative to Figure 1 and corresponding text were carried out;

Reply: Relaxation measurements were performed in 20 mM HEPES buffer (pH 7.2) at 25°C. This information was added in our revised manuscript (lines 94-95 and 98-99 of page 3, including figure caption)

9. how do the relaxivity values obtained for C1 and C2 compare with those reported for similar CAs, e. g. Gd-DTPA?

Reply: Thanks for this comment. T1 relaxivity values (~4.5 mM-1s-1) of the CA1 and CA2 at 60 MHz are similar to one of clinically available MRI T1 contrast agents, Gd-DTPA (Magnevist, Schering, Berlin, Germany), of which T1 relaxivity at 1.5 T (63.8 MHz) is 4.1 mM-1s-1. We added a description on this comparison in Results section (lines 98-102 of page 3) with one additional reference.

10. how do the Authors explain the lack of formation of adducts with HSA, that actually seems to be very likely because of a hydrophobic part in both ligands?

Reply: Actually we don’t know about the exact interaction parts between HSA and our compounds, which may be done in further study.

11. a thorough relaxometric investigation (e. g. the acquisition and fitting of NMRD profiles) is strongly recommended in order to establish important parameters (such as tauM and tauR) that would support some of the conclusions reported by the Authors.

Reply: Thanks for this comment. Main issue of our study is not intrinsic relaxivity of the CAs but it’s increase following interaction with Trx protein. T1 relaxivities of our CAs are similar to clinically used Gd-DTPA which has been well studied on field dependence of T1 relaxivity using NMRD measurements.

Reviewer 2 Report

Manuscript title: An activatable T1w MR contrast agent: noninvasive tool for tracking the vicuna thiol motif of thioredoxin in live cells. 

Authors have developed new T1-sensitive MR contrast agents which enhanced ~ 140% in R1. This sounds quite interesting and would be useful for clinical examinations. I have got few points to be considered by authors.

  1. Please clarify the sequence protocol of the MRI experiment. Are the sequence and TR/TE appropriate to acquire T1w? What is a flip angle? The multislice multi-echo sequence is typically used for T2w?
  2. What RF coil did you use for the MR experiment at 4.7T?
  3. Please discuss further the toxicity of the newly developed contrast agent for clinical use.
  4. Please discuss the feasibility of dynamic contrast enhance imaging using the contrast agent in comparison to the Gd-based method.
  5. How would this agent behave at different field strengths, e.g. 1.5T or 3T? It would be useful to characterise this point.
  6. What about T2? Have you checked if the added contrast agent changes T2 values? I guess not much but we need to ensure it shouldn’t.

Author Response

Reviewer 2

Authors have developed new T1-sensitive MR contrast agents which enhanced ~ 140% in R1. This sounds quite interesting and would be useful for clinical examinations. I have got few points to be considered by authors.

1. Please clarify the sequence protocol of the MRI experiment. Are the sequence and TR/TE appropriate to acquire T1w? What is a flip angle? The multislice multi-echo sequence is typically used for T2w? What RF coil did you use for the MR experiment at 4.7T?

Reply: Thanks for this comment. MR imaging was performed on a 4.7 T MRI system (Biospec 47/40, Bruker, Germany) with a quadrature birdcage RF coil (72 mm diameter) for signal transmission and reception. T1-W images were acquired using MSME (multi-slice multi-echo) MRI pulse sequence with the following parameters: TE/TR = 8.8/400 ms, number of average = 4, matrix size = 192 × 192, FOV = 6 × 3 cm2, slice thickness = 1 mm. We added this description in the Experimental section on lines 298-302 of pages 9-10.

2. Please discuss further the toxicity of the newly developed contrast agent for clinical use.

Reply: We already have done on cell cytotoxicity of MCF cells after incubation with CA1 for 24 hours, and the result is shown in Figure S15, in which there is no cytotoxicity up to 200 μM.

3. Please discuss the feasibility of dynamic contrast enhance imaging using the contrast agent in comparison to the Gd-based method.

Reply: Thanks for this comment. Dynamic contrast enhancement of our CAs is out of current study, so we haven’t done on this issue, however this might be done in further study.

4. How would this agent behave at different field strengths, e.g. 1.5T or 3T? It would be useful to characterise this point.

Reply: Thanks for this comment. T1 relaxivity of our CAs at 1.5 T and 3 T might be similar to Gd-DTPA, however this might be done in further study.

5. What about T2? Have you checked if the added contrast agent changes T2 values? I guess not much but we need to ensure it shouldn’t.

Reply: Thanks for this comment. We are not sure, but T2 relaxivity of the CAs might be similar to clinically used Gd-DTPA. T2 relaxivity of Gd-DTPA at 1.5 T is similar to T1 relaxivity, which is characteristic property of Gd-based T1 contrast agent, resulting in drawback as a T2 contrast agent. Therefore this issue is somewhat far from current study.